# Differential Effects of Yeast NADH Dehydrogenase (Ndi1) Expression on Mitochondrial Function and Inclusion Formation in a Cell Culture Model of Sporadic Parkinson’s Disease

**DOI:** 10.3390/biom9040119

**Published:** 2019-03-27

**Authors:** Emily N. Cronin-Furman, Jennifer Barber-Singh, Kristen E. Bergquist, Takao Yagi, Patricia A. Trimmer

**Affiliations:** 1Neuroscience Graduate Program, University of Virginia, P.O. Box 801392, Charlottesville, VA 22908, USA; enc3p@virginia.edu; 2Parkinson’s and Movement Disorders Center, Virginia Commonwealth University, P.O. Box 980539, Richmond, VA 23298, USA; kristenbergquist@gmail.com; 3Department of Molecular and Experimental Medicine, The Scripps Research Institute, 10550 North Torrey Pines Road, La Jolla, CA 92037, USA; jensingh@gmail.com (J.B.-S.); tycomplex1@gmail.com (T.Y.); 4Department of Anatomy and Neurobiology, Box 980709, Virginia Commonwealth University, Richmond, VA 23298, USA

**Keywords:** trans-mitochondrial cybrid 1, Lewy body 2, mitochondrial dysfunction 3, Ndi1 4, oxygen utilization 5, alpha-synuclein 6

## Abstract

Parkinson’s disease (PD) is a neurodegenerative disorder that exhibits aberrant protein aggregation and mitochondrial dysfunction. Ndi1, the yeast mitochondrial NADH dehydrogenase (complex I) enzyme, is a single subunit, internal matrix-facing protein. Previous studies have shown that Ndi1 expression leads to improved mitochondrial function in models of complex I-mediated mitochondrial dysfunction. The trans-mitochondrial cybrid cell model of PD was created by fusing mitochondrial DNA-depleted SH-SY5Y cells with platelets from a sporadic PD patient. PD cybrid cells reproduce the mitochondrial dysfunction observed in a patient’s brain and periphery and form intracellular, cybrid Lewy bodies comparable to Lewy bodies in PD brain. To improve mitochondrial function and alter the formation of protein aggregates, Ndi1 was expressed in PD cybrid cells and parent SH-SY5Y cells. We observed a dramatic increase in mitochondrial respiration, increased mitochondrial gene expression, and increased PGC-1α gene expression in PD cybrid cells expressing Ndi1. Total cellular aggregated protein content was decreased but Ndi1 expression was insufficient to prevent cybrid Lewy body formation. Ndi1 expression leads to improved mitochondrial function and biogenesis signaling, both processes that could improve neuron survival during disease. However, other aspects of PD pathology such as cybrid Lewy body formation were not reduced. Consequently, resolution of mitochondrial dysfunction alone may not be sufficient to overcome other aspects of PD-related cellular pathology.

## 1. Introduction

Parkinson’s disease (PD) is a neurodegenerative movement disorder characterized clinically by resting tremor, bradykinesia, rigidity, and postural instability [1]. Motor symptoms of PD correlate with the progressive loss of axonal terminals in the striatum and degeneration of dopaminergic neurons in the substantia nigra pars compacta [2]. PD patients also experience non-motor symptoms including autonomic, cognitive and gastrointestinal dysfunction, sleep disturbance, and depression which illustrate the widespread distribution of PD pathology [3]. PD is characterized morphologically by intracellular inclusions called Lewy bodies (perinuclear) and Lewy neurites (inclusions of similar composition located in axons) that form in vulnerable central and peripheral neurons [4]. The etiology of sporadic PD remains unclear although a variety of cellular processes including mitochondrial dysfunction, oxidative stress, aberrant protein aggregation, and impaired autophagy have been strongly implicated [5]. Of these processes, mitochondrial dysfunction has received increasing attention given that neurotoxins, mitochondrial DNA (mtDNA) deletions in PD neurons and several familial gene mutations linked to PD impair mitochondrial function [6,7,8].

Mitochondria utilize oxidative phosphorylation (OXPHOS) to produce ATP from ADP and inorganic phosphate [9]. Optimal mitochondrial function requires the correct assembly and function of the five complexes that comprise the electron transport chain (ETC). Four of the five protein complexes are bigenomic with constituent subunits encoded by nuclear DNA (nDNA) as well as by mtDNA. Complex I, also called nicotinamide adenine dinucleotide (NADH):ubiquinone oxidoreductase, has seven subunits encoded by the mtDNA in addition to the 37 subunits encoded by the nDNA (reviewed by References [10,11]). According to Papa and De Rasmo [10], the intricacy of complex I composition and assembly creates the opportunity for plasticity and makes it an “adaptable regulator of the overall OXPHOS system in mammalian cells”. Consequently, complex I is more vulnerable to detrimental environmental, epigenetic and genetic factors than other complexes in the ETC [10]. Functionally, complex I accepts electrons from NADH at the origin of the ETC and passes the electrons on to complex III via coenzyme Q (CoQ10). Importantly, the dysfunction of complex I has been observed in the brain as well as in peripheral tissues of sporadic and familial PD patients [12,13,14,15]. Furthermore, neurotoxins (ex. rotenone, 1-methyl-4-phenyl-1,2,3,6-tetrahydropyridine, MPTP) that inhibit complex I have been found to induce parkinsonian pathological changes [16,17].

Potential therapies for PD have targeted OXPHOS dysfunction in mitochondria [7,18]. For example, creatine is a naturally occurring compound that is converted to phosphocreatine and serves as an energy source in muscles and nerve cells (reviewed by Pienaar and Chinnery [19]). Creatine is neuroprotective in a wide range of cellular and animal models of PD [19,20]. A recent evaluation of five randomly controlled clinical trials using creatine to treat PD patients concluded that there was no clinical benefit although more correlated studies are still needed [20,21]. CoQ10, a key electron carrier for the ETC, has also been shown to act as an antioxidant and to provide neuroprotection in many PD model systems [20]. In a recent meta-analysis, CoQ10 was shown to be safe and well tolerated, but this therapy displayed no evidence of symptomatic benefit or delay in functional decline [22]. MitoQ, a mitochondrial-targeted CoQ10 was developed and has also been explored as a potential mitochondrial therapy in various animal models of mitochondrial dysfunction [19,23]. Since CoQ10 and MitoQ exhibited limited success in PD, other compounds and approaches are being designed to target mitochondrial dysfunction [18,24,25]. In general, studies in PD patients have not yet shown that these and other neuroprotective compounds alone can consistently and effectively improve PD symptoms or slow disease progression [7].

An alternate approach that has been pursued in a variety of PD models is to supplement dysfunctional complex I in mitochondria in order to improve ETC function [26,27]. The yeast strain *Saccharomyces cerevisiae* contains both internal and external single subunit NADH dehydrogenase enzymes. The internal NADH dehydrogenase (Ndi1) faces the mitochondrial matrix where NADH is formed via the Krebs cycle [28]. Similar to mammalian complex I, Ndi1 accepts electrons from NADH and passes them on to CoQ10. However unlike mammalian complex I, Ndi1 is a single, monogenic protein encoded by the *NDI1* gene and is insensitive to the complex I inhibitor rotenone [29].

Ndi1 expression studies thus far have shown that it becomes localized to mitochondria where it is functionally active, does not induce an inflammatory or immune response, and is well tolerated by mammalian cells [30,31,32,33,34,35,36]. Cell growth and viability in human or rodent cell lines, such as HEK293, 143B, and PC12 cells were not adversely impacted by trans-species expression of Ndi1 [31,33,37]. In fact, Ndi1 expression improved OXPHOS capacity and restored NADH oxidase activity in Mdivi-1 expressing COS-7 cells [38], complex I deficient human 143B osteosarcoma cells [32,39], and a 143B cell model of Leber’s hereditary optic neuropathy (LHON) bearing the G11778A mutation in the ND4 subunit of complex I [30]. Comparable improvements were also seen in animal models of LHON and in a model of defective complex I assembly in *Drosophila* [40,41,42]. Furthermore, Ndi1 was ubiquitously expressed to create a transgenic *Drosophilia* [43]. In this model, Ndi1 expression rescued flies from the knockdown of complex 1, reduced age-related decline in respiratory function, and increased life span. More importantly, in cell and animal neurotoxin models of PD such as rotenone and MPTP, Ndi1 expression reduced neuronal cell death and oxidative damage and minimized behavioral changes [26,33,37,44,45,46,47,48]. Ndi1 expression effectively supplemented dysfunctional complex I irrespective of the location of the defective subunit gene (nuclear or mitochondrial) [40,49]. In light of these studies, we expressed Ndi1 in a human cybrid cell culture model of sporadic PD that exhibits a deficit in ETC assembly and function and the formation of LB-like inclusions [50] to determine if improved OXPHOS after Ndi1 expression is linked to improvements in other PD-related cellular pathology.

PD cybrid (cytoplasmic hybrid) cell lines were created by fusing mtDNA-depleted SH-SY5Y human neuroblastoma cells (rho0) with platelets from an individual diagnosed with sporadic PD [12,50]. Changes in platelet biomarkers correlate with PD progression and have been shown to be predictive for Alzheimer’s disease (AD) and cognitive decline [51,52]. Cybrids made from PD, AD, multiple sclerosis, and mild cognitive impairment patients also model changes seen in subjects’ brain tissue [53,54,55,56,57]. While the resulting cybrid lines express mtDNA from an individual PD patient, all the lines share the same SH-SY5Y nuclear background and environmental conditions in culture [58].

The PD cybrid cell lines in our lab exhibit heterogeneous mitochondrial haplotypes and phenotypes due to mtDNA contributed by each PD patient’s platelets [58,59]. We specifically selected a sporadic PD cybrid cell line (PD61) that has typical ETC dysfunction with reduced complex I assembly, expression and function, to use for this extensive study [58,59]. The platelets used to generate PD61 were donated by a 65-year-old male 15 years after diagnosis (Hoehn and Yahr stage 2). According to Pignataro et al. [58], the haplotype of PD61 is L2e1a (sub Saharan). In addition to the mutations characteristic of this haplogroup, PD61 contains six additional individual coding-region mutations [58]. PD61 also exhibits reduced mtDNA gene expression, copy number, and spontaneously forms intracellular, perinuclear aggregates called cybrid Lewy bodies (CLB) that replicate the composition and structure of cortical Lewy bodies (LB) [50,59,60]. Constituents of CLB include alpha-synuclein (αSYN), ubiquitin and other components of the ubiquitin proteasome system, lysosomes, Congo red- and thioflavin S-staining, proteins damaged by oxidation and nitration and mitochondrial components (see References [50,60]). By using a CLB-expressing PD cybrid line for this study, we also planned to investigate if complementation of mitochondrial dysfunction by Ndi1 expression would impact on levels of αSYN, small protein aggregates and CLB. The parent SH-SY5Y cell line was also transfected with Ndi1 as a control and to explore the consequences of Ndi1 expression for cells with intact mitochondrial ETC function.

Our results show that Ndi1 expression in a human cybrid model of sporadic PD significantly improved oxygen utilization, levels of protein aggregates and some abnormalities in OXPHOS but did not alter the expression levels of CLB. These data suggest that restoration of mitochondrial ETC function by circumventing complex I dysfunction through Ndi1 expression provides only a partial restoration of cellular and mitochondrial function. Additional strategies will be needed to fully compensate for other cellular dysfunctions that play a role in the pathogenesis of sporadic PD.

## 2. Materials and Methods

### 2.1. Generation of PD Cybrids and Cell Culture

As described by Swerdlow et al. [12], rho0 SH-SY5Y cells were fused with platelets isolated from aseptically-drawn PD patient blood samples by brief incubation with polyethylene glycol (J.T. Baker, Thermo Fisher Scientific, Waltham, MA, USA). Residual rho0 cells and cybrid cells with insufficient platelet mitochondria repopulation were eliminated by growth in cybrid selection medium containing high glucose Dulbecco’s Modified Eagle Medium (DMEM, Life Technologies, Thermo Fisher Scientific, Waltham, MA, USA), dialyzed fetal calf serum (Hyclone/Thermo Fisher Scientific, Waltham, MA, USA), antibiotic/antimycotic, and lacking pyruvate and uridine for 5–6 weeks [12,50]. Cybrid cell lines that tested negative for mycoplasma were aliquoted and stored in liquid nitrogen.

All cell lines were cultured in T75 cm^2^ flasks (Greiner Bio-One, Monroe, NC, USA) with growth medium consisting of DMEM supplemented with 10% fetal bovine serum (FBS, Hyclone/Thermo Fisher, Waltham, MA, USA), pyruvate/uridine, and antibiotic/antimycotic as previously described [60]. Ndi1 selection media was made with glucose-free DMEM, supplemented with 10% FBS, pyruvate/uridine, antibiotic/antimycotic, 5 mM galactose, and 30 nM rotenone [60]. Rotenone was prepared in advance as a stock and added directly to the media during the selection period. Ndi1-transfected cell lines were provided by Yagi Lab at passage 4–5. Cells used for the studies reported here were at passage 12–20. Ndi1-expressing cell lines were maintained in growth medium but were returned to selection media every 5 days for 48 h to maintain a stable Ndi1 expression level. To minimize the chance of mycoplasma infection, all cell lines were discarded after 2 months and replaced by fresh cells from frozen aliquots.

### 2.2. Viral Transfection

Gene delivery by adeno-associated virus (AAV) is safe, effective and elicits long-lasting expression [44,45]. Cells were transfected with the rAAV-*NDI1* (serotype 2), as previously described [33]. Briefly, PD61 and SH-SY5Y cells were plated for 48 h in 6-well plates before addition of the virus. Viral transfection used 5–8 × 10^12^ virus particles/mL of growth medium to ensure efficient transfection [37]. Cells were incubated with the virus for 5 days, then returned to regular growth medium and allowed to proliferate. Transfected cell populations were then exposed to selection media for at least 2 weeks. Immunocytochemistry with a Ndi1-specific antibody was used to verify that the selection process was complete [26]. Ndi1-expressing cell lines were removed from selection media before preparation of frozen stocks.

### 2.3. Immunocytochemistry

Cells were plated in polylysine-coated 35-mm coverslip dishes (#0 thickness coverslips, Mat-Tek Corporation, Ashland, MA, USA), grown for 72–96 h (75% confluency) and then fixed with 4% paraformaldehyde in phosphate buffered saline (PBS). Cells were either incubated in citrate antigen retrieval buffer (10 mM, 95 °C, 20 min) and blocked with 1%BSA/1%Triton/PBS blocking buffer for 15 min (for Ndi1, LAMP2A, and αSYN) or permeabilized with 5% urea antigen retrieval buffer (pH 9.5, 95 °C, 20 min) followed by 0.2% Triton and blocked with 10% goat serum in PBS (30 min for Complex Vα and porin). Cells were incubated with primary antibodies overnight at 4 °C and stained with secondary antibodies conjugated with a fluorophore (AlexaFluor 488, #568; 1:400 (Thermo Fisher Scientific, Waltham, MA, USA) or MS602-IgG_2b_-Texas Red, 1:200, Mitosciences/abcam, Cambridge, MA, USA). Antibodies used were: Ndi1 (1:600, provided by T. Yagi), Lamp2A, (1:200, #555803, BD Biosciences USA, San Jose, CA, USA), αSYN (1:400, #AB5038, Chemicon/Thermo Fisher Scientific, Waltham, MA, USA), Complex Vα, and porin (both 1:400, Mitosciences/Abcam, Cambridge, MA, USA).

### 2.4. Immunoblots

Cells were grown in T175 cm^2^ flasks (Greiner Bio-One, Monroe, NC, USA) to 80–90% confluency and harvested in 1X radioimmunoprecipitation assay buffer with protease inhibitors and phenylmethanesulfonylfluoride as previously described [59]. The soluble fraction was isolated by centrifugation. Protein quantity was measured using the Micro BCA kit (Pierce/Thermo Fisher Scientific, Waltham, MA, USA). Equal amounts of protein were loaded on to Bis-Tris gels (Bio-Rad, Hercules, CA, USA) and run using the Bio-Rad Criterion system. Proteins were transferred onto nitrocellulose membranes using the iBlot transfer system (Thermo Fisher Scientific, Waltham, MA, USA). Membranes were blocked using Li-Cor blocking buffer [26] and stained with primary antibodies to Ndi1 (1:2000, provided by T. Yagi), MitoProfile Total OXPHOS cocktail, (1:400, Mitosciences/abcam, Cambridge, MA, USA) or αSYN (sc-7011R, 1:100, Santa Cruz Biotechnology, Dallas, TX, USA) at room temperature. Membranes were then washed and stained at room temperature with appropriate secondary antibodies labeled with an infrared dye (1:4000, Li-Cor, Lincoln, NE, USA). Membranes were imaged using the Odyssey scanner (Li-Cor, Lincoln, NE, USA). Band densities were calculated as integrated intensities using the Odyssey software. Integrated intensities were normalized to porin (MSA03, 1:2000, Mitosciences/Thermo Fisher Scientific, Waltham, MA, USA) or actin (A2103, 1:2000, Millipore-Sigma, Burlington, MA, USA). Student’s *t*-test was used to compare normalized integrated intensities between non-transfected and Ndi1 transfected cells, with Welch’s correction for unequal variances when necessary (Prism, GraphPad, San Diego, CA, USA).

### 2.5. Quantitative Real-Time PCR (qRT-PCR)

Pellets of approximately 10 × 10^6^ cells were collected from sub-confluent T175 cm^2^ flasks. Total RNA and DNA were isolated using an RNA/DNA isolation kit (Qiagen, Germantown, MD, USA) and measured using a NanoDrop (Thermo Fisher). From RNA, complementary DNA (cDNA) was made using the iScript cDNA synthesis kit (Bio-Rad, Hercules, CA, USA). Quantitative real-time PCR was run for single genes (EvaGreen Power mix, Bio-Rad, Hercules, CA, USA) or in a multiplex set (iQ Multiplex Power mix, Bio-Rad) in a CFX96 Real-Time PCR Detection System (Bio-Rad, Hercules, CA, USA). Bio-Rad CFX Manager software calculated starting quantities for samples, based on a standard curve with known starting quantities. All values were normalized to endogenous reference genes based on geNorm (BioGazelle, Technologiepark 3 Zwijnaarde, 9052, Belgium) analysis to find the genes with the highest expression stability. For gene expression, we used the endogenous reference genes for glyceraldehyde 3-phosphate dehydrogenase and 14-3-3-zeta. We used 14-3-3-zeta and beta-2-microglobulin as the endogenous reference genes for DNA copy number studies. Student’s *t*-tests were used to analyze data (Prism, GraphPad, San Diego, CA, USA).

### 2.6. Cellular Respiration (OXPHOS)

Cellular respiration was measured using the Seahorse Extracellular Flux Analyzer XF24 (Seahorse Bioscience/Agilent, Santa Clara, CA, USA) as previously described [60]. Cells were grown to a confluent monolayer in XF24 culture plates for 24 h prior to the assay. Mitochondrial inhibitors used were oligomycin (1 μM) to inhibit ATP synthase, carbonyl cyanide 4-(trifluoromethyoxy) phenhydrazone (FCCP, 300 nM) to dissipate the inner mitochondrial membrane proton gradient, rotenone (100 nM) to inhibit complex I, and antimycin A (10 μM) to inhibit complex III, all at a pH of 7.4. All measurements were acquired after a 3-min mix and 2-min waiting period to re-oxygenate the media. Oxygen concentration was then measured in sequential 3-min windows in order to calculate an oxygen consumption rate (OCR). Prism software was used to determine basal, maximum capacity and non-mitochondrial OCR as well as to calculate leak, ATP-linked OCR and spare capacity [61]. The basal extracellular acidification rate (ECAR) was also determined. Following each experiment, individual OCR and ECAR values were normalized to protein content (Micro BCA Kit, Pierce/Thermo Fisher Scientific, Waltham, MA, USA). The contribution of Ndi1 expression to the OCR measurements for each cell line was measured by inhibiting endogenous complex I (30 nM rotenone in the running media starting 1 h prior to initiating the XF24 experiment-Pre-RTN). Statistics were calculated using two-way ANOVA with Bonferroni multiple comparison post-hoc tests. (Prism, GraphPad, San Diego, CA, USA).

### 2.7. Aggregated Protein and CLB Measurements

Ndi1 transfected and non-transfected cell lines were stained with 100 mM Congo red for 24 h to visualize beta-pleated sheet aggregated proteins and CLB as previously described [60]. The density of Congo red aggregates (Congo red positive pixels) was quantified using MetaMorph image analysis software (Molecular Devices, San Jose, CA, USA) and then normalized to cell number per each image. CLB, defined as spherical Congo red positive inclusions over 2 μm in diameter, were visualized using a FV1000 laser scanning confocal microscope, counted and measured using Fluoview software (Olympus America, Center Valley, PA, USA). Experiments were repeated with cells from a different passage. Outcomes from transfected and non-transfected cell lines were compared using Student’s *t*-test (Prism, Graph Pad, San Diego, CA, USA).

### 2.8. Mitochondrial Movement

Transfected and non-transfected cell lines were differentiated in 35mm polyethyleneimine-coated dishes as previously described [60,62]. Undifferentiated cells were grown for 24–48 h in regular growth media before differentiation. Differentiation media consisted of Neurobasal medium (Thermo Fisher Scientific, Waltham, MA, USA), B27 supplement (Thermo Fisher Scientific, Waltham, MA, USA), antibiotic/antimycotic, pyruvate/uridine and freshly diluted staurosporine (4–6 nM, Millipore-Sigma, Burlington, MA, USA) as previously described [60]. Transfected and non-transfected cell lines were maintained in differentiation media for 12 days. To image mitochondrial movement in differentiated neurites, transfected and non-transfected neuronal cells were incubated with MitoTracker CMXRos (MitoTrackerRed, Molecular Probes, Thermo Fisher Scientific, Waltham, MA, USA) as previously described [60,63]. Time-lapse recordings of fluorescently labeled mitochondria were collected using MetaMorph Imaging software (Molecular Devices, San Jose, CA, USA) every 3 s for 2 min using an Olympus IX70 inverted microscope equipped with epifluorescence and Nomarski optics, a Lambda 10-2 filter wheel, a Photometrics CoolSnap HQ progressive scan CCD camera, and a heater/controller to maintain cybrid cells at 37 °C during image collection (World Precision Instruments, Inc., Sarasota, FL, USA). All the mitochondria were individually tracked in each movie frame in randomly selected processes (PD61 n = 3 cultures, 51 processes; PD61^Ndi1^ n = 4 cultures, 49 processes; SH-SY5Y n = 3 cultures, 53 processes; SH-SY5Y^Ndi1^ n = 3 cultures, 49 processes). Mitochondrial velocities were calculated using MetaMorph Imaging Software and analyzed using the Students *t*-test.

## 3. Results

### 3.1. Ndi1 Expression in PD61 Cybrid and SH-SY5Y Cell Lines

Successful transfection and selection of PD61 and SH-SY5Y cell lines for expression of rAAV-NDI1 (PD61^Ndi1^ and SH-SY5Y^Ndi1^) was established using immunocytochemistry, qRT-PCR and immunoblots (Figure 1). Levels of Ndi1 protein as well as gene expression were undetectable in non-transfected PD61 and SH-SY5Y cell lines (data not shown). The incorporation of Ndi1 protein into the mitochondria was confirmed by co-localization of Ndi1 immunostaining with immunostaining for complex Vα (Figure 1A,B). Ndi1 protein level in SH-SY5Y^Ndi1^ cells was significantly increased (Figure 1C,D). In contrast, Ndi1 protein expression levels were six-fold higher in PD61^Ndi1^ than in SH-SY5Y^Ndi1^ (Figure 1C). Similarly, using qRT-PCR, *NDI1* gene expression was three-fold higher in PD61^Ndi1^ than in SH-SY5Y^Ndi1^ (Figure 1D).

### 3.2. Effects of Ndi1 Expression on Oxygen Consumption Rates (OCR) and Extracellular Acidification Rates (ECAR)

Oxygen utilization in non-transfected and Ndi1-expressing cell lines was assessed using the Seahorse XF24 (Seahorse Bioscience/Agilent) [61,64]. The OCR and ECAR (a surrogate measure of glycolysis) were calculated using XF24 Analyzer software (see Methods). Sequential injection of inhibitors permits the measurement of specific components of the ETC such as basal, ATP-linked and maximum capacity, non-mitochondrial OCR, and calculation of spare capacity and leak as shown in Figure 2A and Figure 3A (see [60,61]). ATP-linked OCR was determined after the addition of oligomycin to inhibit ATP synthesis (Figure 2A and Figure 3A). The difference between basal and oligomycin OCR is the oxygen utilization that is ATP-linked. Leak is the OCR remaining after oligomycin (adjusted for non-mitochondrial oxygen utilization, see Figure 2A and Figure 3A). Maximum OCR is determined after the addition of FCCP (Figure 2A and Figure 3A) and spare or reserve capacity is calculated as the difference between maximum and basal OCR.

### 3.3. PD61/PD61^Ndi1^

Figure 2A shows a typical bioenergetics profile for PD61 and PD61^Ndi1^. Basal OCR (net sum of all oxygen consuming processes at baseline) in PD61 was low (2082 pmolO_2_/min/mg total protein) while basal OCR of SH-SY5Y cells was 20% higher at 11,796 pmolO_2_/min/mg total protein (compare Figure 2C and Figure 3C). Values for ATP-linked, leak, maximum, spare capacity, and non-mitochondrial OCR were also minimal in PD61 (Figure 2C). The presence of Ndi1 protein in PD61 had a robust effect on respiration, resulting in a 10-fold increase in basal OCR levels (Figure 2A,C). As shown in Figure 2C, PD61^Ndi1^ also exhibited a considerable and significant increase in ATP-linked, maximum capacity and non-mitochondrial OCR (not shown) as well as spare capacity and leak. OCR was also measured for four control cybrid cell lines generated from disease-free, age-appropriate donor platelets. The basal OCR for PD61^Ndi1^ was 2.1X higher than the mean basal OCR for the four control (CNT) lines (21,955 pmolO_2_/min/mg total protein for PD61^Ndi1^ as compared to 10,311 pmolO_2_/min/mg total protein for CNT, *p* < 0.001, 1-way ANOVA). Similarly, the max capacity OCR in PD61^Ndi1^ also exceeded the max capacity for the four control lines by 2.4X (30,768 vs. 13,028 pmolO_2_/min/mg total protein, *p* < 0.0001, 1-way ANOVA).

Basal ECAR was higher in PD61 compared to SH-SY5Y (compare Figure 2B and Figure 3B). This is consistent with other studies showing that cells with compromised oxygen utilization maintain ATP levels by increased glycolysis [65]. PD61^Ndi1^ exhibited a significant increase in basal OCR that was accompanied by a significant 50% decrease in ECAR in PD61^Ndi1^ (Figure 2B) to become more reliant on OXPHOS than on glycolysis.

To confirm that Ndi1 was functionally contributing to the changes seen in OXPHOS in PD61^Ndi1^ (Figure 2E), we measured OCR in the presence of 30 nM rotenone (rotenone pretreatment or Pre-RTN, Figure 2E). Under these conditions, rotenone will inhibit endogenous complex I but will not affect the activity of rotenone-resistant Ndi1. In spite of the poor oxygen utilization in PD61, pretreatment with rotenone significantly decreased endogenous basal, ATP-linked, maximum, and spare capacity OCR in PD61 (Figure 2D). Rotenone pretreatment of PD61^Ndi1^ (Figure 2E) significantly reduced but did not eliminate oxygen utilization. Since Ndi1 is not inhibited by rotenone, the respiration remaining after pretreatment with rotenone is due to the expression of Ndi1 in PD61 mitochondria. The reductions in PD61^Ndi1^ in basal, leak, maximum and spare capacity OCR were all significant when pre-treated with rotenone (Figure 2E). However, levels of basal, leak, maximum, and spare capacity OCR remained high even after rotenone pre-treatment (Figure 2E).

### 3.4. SH-SY5Y/SH-SY5Y^Ndi1^

Figure 3A shows the bioenergetic profile of SH-SY5Y and SH-SY5Y^Ndi1^. Ndi1 expression in SH-SY5Y cells had no significant effect on basal, ATP-linked or leak OCR (Figure 3C). However maximum and spare capacity OCR levels were significantly increased in SH-SY5Y^Ndi1^ (Figure 3C). Even though basal OCR levels were unchanged in SH-SY5Y^Ndi1^, there was a significant 55% decrease in ECAR in SH-SY5Y^Ndi1^ compared to SH-SY5Y (Figure 3B) indicating a shift toward greater reliance on OXPHOS rather than glycolysis.

Pretreatment with rotenone (Figure 3D) to eliminate endogenous complex I activity resulted in significant reductions in basal, ATP-linked and maximum capacity OCR in non-transfected SH-SY5Y cells that virtually eliminated oxygen utilization. In comparison, rotenone pre-treatment in SH-SY5Y^Ndi1^ (Figure 2E) had no effect on basal, ATP-linked or leak OCR. Rotenone pretreatment of SH-SY5Y^Ndi1^ resulted in a significant decline only in maximum and spare capacity OCR.

### 3.5. Effects of Ndi1 Expression on Mitochondrial Movement

Previous studies have shown reduced velocities of mitochondrial movement in PD cybrid neurites compared to SH-SY5Y [60,63]. Mitochondrial movement was measured in Ndi1-expressing and non-transfected, differentiated PD61 and SH-SY5Y neurites. The velocities of all the mitochondria in each neurite were averaged and then graphed to show the range of mitochondrial velocities (Figure 4A,B). The average velocity of mitochondria in all PD61^Ndi1^ neurites was not significantly different from the average mitochondrial velocity in PD61. However, a histogram analysis to visualize average mitochondrial velocity per neurite revealed that Ndi1 expression resulted in a shift from slow (0.04 to 0.20 microns/second) to more rapid mitochondrial movement (0.21 to 0.44 microns/second) in PD61^Ndi1^ (Figure 4B). Furthermore, the percentage of mitochondria classified as “not moving” (velocity ≤ 0.05 microns/second) was significantly reduced in (*p* = 0.03, matched *t*-test) in PD61^Ndi1^ neurites compared to PD61 neurites (5.7% ± 1.7 SEM for PD61^Ndi1^ vs. 13.2% ± 2.2 SEM for PD61). Also, the percentage of mitochondria that moved for less than 9 consecutive seconds was significantly reduced (*p* = 0.025 matched *t*-test) in PD61^Ndi1^ (23.3% ± 5.3 SEM) compared to PD61 (36.8% ± 3.5 SEM).

Mean mitochondrial velocities for SH-SY5Y and SH-SY5Y^Ndi1^ per neurite were not significantly different based on a similar histogram analysis as described above. There were no significant changes in the percentage of non-moving mitochondria or in the percentage of mitochondria that moved for less than nine consecutive seconds in SH-SY5Y vs. SH-SY5Y ^Ndi1^ (data not shown).

PD61 and PD61^Ndi1^ cells successfully differentiated into neurons with multiple long processes that resemble the axons (long, unbranched, narrow caliber) and dendrites (tapering and branched) similar to those formed by SH-SY5Y neurons (not shown). In contrast, many SH-SY5Y^Ndi1^ cells produced shorter processes in response to the low dose of staurosporine differentiation. Further study is needed to determine if the altered appearance of neurites in SH-SY5Y^Ndi1^ is a consequence of Ndi1 expression in this neuroblastoma cell line or if the differentiation protocol requires modification for SH-SY5Y^Ndi1^. Previous studies using transfected PC12 cells showed that Ndi1 expression did not adversely affect the morphology of neuronal cells that were already differentiated [37].

### 3.6. Effects of Ndi1 Expression on Endogenous Mitochondrial Gene Expression

We measured gene expression and copy number of four genes encoded by mtDNA- ND2 and ND4 (complex 1 subunits), the complex IV subunit (cytochrome oxidase) COX3, and the 12s ribosomal RNA (rRNA). The copy number of all four genes in PD61 was significantly below the SH-SY5Y level (Dashed line at 1.0 in Figure 5A). Mitochondrial gene expression for ND2, ND4, and COX3 but not 12s rRNA was significantly increased in PD61^Ndi1^ compared to PD61 (Figure 5A). In fact, Ndi1 bypass of complex I dysfunction in PD61^Ndi1^ resulted in mtDNA gene copy numbers that were significantly increased to levels above SH-SY5Y gene expression levels (Dashed line at 1.0 in Figure 5C,D). Gene expression for ND2, ND4, COX3 and 12s rRNA was also increased in SH-SY5Y^Ndi1^, however, these changes did not achieve significance (Figure 5B). mtDNA copy number also did not change significantly in SH-SY5Y^Ndi1^ compared to SH-SY5Y (Figure 5D).

### 3.7. Effects of Ndi1 Expression on Mitochondrial Biogenesis

Peroxisome proliferator-activator receptor gamma co-activator 1-alpha (PCG-1α) is a transcription factor that serves as the master regulator for mitochondrial biogenesis [66]. In both cell lines expressing Ndi1, we observed a significant increase in PGC-1α expression compared to the non-transfected cell lines (Figure 6A). PD61^Ndi1^ had the highest relative expression of PGC-1α, which was not surprising given the robust changes in mitochondrial function and mtDNA gene expression. Nuclear respiratory factor 1 (NRF1) is a DNA binding protein that regulates nuclear encoded ETC subunits as well as mitochondrial transcription factor B1 and mitochondrial transcription factor A (TFAM) which both bind to mtDNA, initiate mitochondrial-encoded gene transcription, and also participate in mitochondrial biogenesis [67,68,69]. Neither PD61^Ndi1^ or SH-SY5Y^Ndi1^ exhibited significant changes in these mitochondrial transcription factors (not shown).

### 3.8. Effects of Ndi1 Expression on ETC Assembly

In light of the increases in mitochondrial gene expression and biogenesis signaling, we also measured the assembly of the individual ETC complexes using an antibody cocktail (MitoProfile Total OXPHOS human WB antibody cocktail, abcam, Cambridge, MA, USA) that recognizes subunits that are labile when each ETC complex is incorrectly assembled [70]. The assembly of complex III and complex V in PD61^Ndi1^ was significantly increased compared to PD61 (Figure 6C). Levels of assembled complexes I and IV in PD61^Ndi1^ were not significantly different from PD61 (Figure 6C) despite the observation that mtDNA gene expression of subunits for complexes I and IV was significantly increased (Figure 5A). In SH-SY5Y^Ndi1^, levels of assembled complexes I, II, III, and IV were all increased significantly compared to SH-SY5Y (Figure 6D). In comparison, mtDNA gene expression of complex I and IV subunits were not significantly changed by Ndi1 expression in SH-SY5Y (Figure 5B).

### 3.9. Effects on Ndi1 Expression on αSYN

We measured αSYN protein levels using immunoblots because αSYN has been shown to permeabilize mitochondrial membranes and to play a role in the formation of LB [71,72,73]. There was no detectable difference in the quantity of soluble αSYN in Ndi1 expressing cell lines compared to non-transfected lines (data not shown). We also assessed αSYN mRNA levels using qRT-PCR. As shown in Figure 7B, αSYN gene expression was increased almost two-fold in PD61^Ndi1^ versus PD61 but this change did not achieve statistical significance (*p* = 0.055). There was also no significant difference in αSYN gene expression between SH-SY5Y^Ndi1^ and SH-SY5Y (Figure 6B).

### 3.10. Effects of Ndi1 Expression on Levels of Protein Aggregates and CLB

We determined the total cellular level of aggregated proteins, as well as the frequency of CLB expression (see Methods) using live-cell staining with Congo red, a histochemical dye that binds to proteins containing beta-pleated sheet configurations as well as to the N-terminal aggregation prone region of αSYN [60,74,75]. In PD61^Ndi1^, total cellular aggregated protein content (Congo red positive pixels/cell) was significantly decreased, compared with PD61 (Figure 7A). A similar change was observed in SH-SY5Y^Ndi1^ compared to SH-SY5Y (Figure 7A).

Expression levels of Congo red-labeled CLB (round inclusions greater than 2μm in diameter) were also unchanged in PD61^Ndi1^ (1–2% in both PD61 and PD61^Ndi1^, Figure 8B) despite a significant improvement in OXPHOS function and a reduction in the levels of Congo red positive protein aggregates. There was also no change in size of CLB in PD61^Ndi1^ compared to PD61 (4.336 ± 0.5902 for PD61 vs. 4.217 ± 0.938 for PD61^Ndi1^). As shown in Figure 8C,D, CLB continued to be expressed in transfected cells with robust Ndi1 expression. Furthermore, Ndi1 immunostaining was heterogeneously incorporated into CLB. In some cases, CLB contained Ndi1 immunostaining that co-localized with staining for the mitochondrial outer membrane protein porin (Figure 8D).

## 4. Discussion

The yeast mitochondrial NADH dehydrogenase, Ndi1, has been widely used to bypass complex I dysfunction in vitro and in vivo. We expanded on existing literature to investigate Ndi1 expression in a human cell culture model of sporadic PD that exhibits critical aspects of PD pathology including compromised mitochondrial oxygen utilization, the expression of aggregated proteins, and Lewy body-like CLB. Similar to previous studies, we found that trans-species expression of a nuclear-encoded yeast gene NDI1 in our human, cell culture model of sporadic PD compensated for complex I dysfunction, restored NADH oxidase activity, and enabled more efficient operation of the mitochondrial ETC (e.g., References [44,49,76]). While the presence of Ndi1 in PD61 resulted in reduced levels of Congo red-positive aggregated proteins, CLB expression levels and αSYN gene expression were not significantly changed. This suggests that bypass of complex I dysfunction by Ndi1 expression in a model of sporadic PD was sufficient to improve mitochondrial function but was not sufficient to reduce other cellular dysfunctions relevant to the formation and maintenance of CLB.

One of the compelling reasons to explore an alternate NADH oxidase like Ndi1 for PD is that it can overcome OXPHOS dysfunction without regard to the cause of the complex I dysfunction. For example, NDI1 gene expression has been successfully used in a range of models from *Caenorhabditis elegans* to vertebrates [77,78], primary and tumor cell cultures where complex I dysfunction is due to neurotoxicity [17,36,38,44,46], nuclear gene mutation [41,49], or mitochondrial gene mutation [32,40]. Whether functional compromise is the result of complex I inhibition or altered structure, *NDI1* gene expression improved mitochondrial function. Furthermore, Ndi1, like other alternate oxidases from lower organisms, appears to be inactive when cells function normally and only becomes active under conditions when the ETC is dysfunctional [79]. Rustin, Jacobs and the Alternatives Consortium [79] suggest that alternative respiratory enzymes like Ndi1 confer “significant flexibility” to the ETC allowing it to “overcome potential constraints”.

### 4.1. Cellular Consequences of Ndi1 Expression in PD61

We selected PD61 as a test case because the complex I activity and ETC dysfunction were comparable to PD brain [59,80]. Platelets for the creation of the cybrid came from a 65-year-old male with a Hoehn and Yahr score of 2 and disease duration of 15 years. The donor was haplogroup L2e1a (sub-Saharan) with diagnostic mutations as well as six mutations that were unique to the individual relative to the revised Cambridge Reference sequence [58]. Examination of the substantial changes in cellular functions induced by Ndi1 expression including improved oxygen utilization, axonal transport of mitochondria, mitochondrial gene expression, and copy number as well as increased mitochondrial biogenesis signaling by PD61^Ndi1^ could provide insights for future PD therapy development.

Previous studies have shown that oxygen utilization in PD61 is crippled by ETC damage [59,63]. Our measurements of oxygen utilization using the XF24 suggest that increased NADH oxidation due to Ndi1 expression was enough to robustly alter the respiratory profile of PD61^Ndi1^. Although we expected some functional improvements, we were surprised by the magnitude of the respiratory improvements in PD61^Ndi1^. Immunoblot analysis confirmed that porin expression was unchanged by the presence of Ndi1 (Figure 6B) so this change was not due to increased mitochondrial mass. The increased basal, ATP-linked, maximum, and spare capacity OCR in PD61^Ndi1^ is consistent with a significant increase in the volume of electrons being transferred along the ETC.

The ATP-linked OCR of PD61^Ndi1^ was significantly improved compared to PD61 but a greater increase was seen in maximum capacity. Desler et al. [81] suggested that complex IV, the final electron acceptor of the ETC, balances the activity of the ETC in response to cell needs. In support of this proposal, PD61^Ndi1^ had increased expression and gene copy number of COX3, a mtDNA encoded complex IV subunit (Figure 5). Non-mitochondrial OCR was also significantly increased in PD61^Ndi1^ compared to PD61 (not shown, *p* < 0.05). Increased availability of NADH due to Ndi1 expression was available for use by the ETC as well as non-mitochondrial NADPH oxidases accounting for the increase in non-mitochondrial oxygen utilization [82,83].

Since the presence of Ndi1 increases electron transfer along the ETC, electron leak also increased. Electron leak can result in increased free radical generation at complexes I and III [82]. Qualitative assessment of ROS generation by PD61^Ndi1^ using DCFDA (2′,7′-Dichlorodihydrofluorescein diacetate) was not appreciably different from PD61 (data not shown). Lack of increased free radical generation after Ndi1 expression is, however, consistent with Ndi1 studies in other systems [36]. The increased expression of PGC-1α seen in PD61^Ndi1^ could also contribute to suppression of ROS through its role in ROS detoxification [84].

Spare capacity is critically important to synaptic function and neuronal survival since decreases in spare capacity increase neuronal vulnerability and can lead to an energy crisis when ATP demands are increased [81,85]. PD61^Ndi1^ exhibited a significant increase in spare capacity compared to PD61 suggesting that PD61^Ndi1^ cells have increased reserves and are better equipped to handle respiratory stress like that induced by MPTP or rotenone.

The significant increase in OCR we observed in PD61^Ndi1^ was matched by a compensatory 56% decrease in ECAR (a surrogate for glycolysis). Mitochondrial oxygen utilization (OCR) can be significantly reduced by high levels of glucose in culture media that are needed to support dysfunctional PD cybrid cells [86]. SH-SY5Y neuroblastoma and cybrid cell lines exhibited the Warburg effect (low OCR and high ECAR) [87]. Ndi1 expression in PD cybrid cells also reduced their reliance on glycolysis and improved oxygen utilization and facilitated ATP production.

To determine how increased respiratory function after Ndi1 expression alters mtDNA genes, we measured changes in mtDNA gene expression and copy number. Reduced expression of mitochondrial genes has been observed in previous studies of PD human brain, peripheral cells, and cybrids [59,60,80,88]. In keeping with the changes in OCR, we also detected increased levels of mitochondrial gene expression and copy number for subunits of complex I, IV and 12s rRNA in PD61^Ndi1^ compared to PD61.

Increased copy number for complex I genes did not result in an increase in assembled complex I based on immunoblots (Figure 6C). We did not expect complex I assembly to increase since Ndi1 expression only bypasses dysfunctional endogenous complex I and does not restore the production of normal constituent subunits, assembly or normal function. Increased assembly of complexes III and V in PD61^Ndi1^ compared to PD61 is a reasonable outcome since Ndi1 feeds electrons into complex III driving increased ATP-linked respiration. However, increases in mtDNA-encoded complex I and IV subunits (Figure 5) did not translate into increased assembly of ETC complexes (Figure 6). A recent study found subunits that are not immediately incorporated into assembled complexes may be quickly degraded [89]. This could explain why an increase in mtDNA gene expression was observed without a consistent effect on ETC complex assembly.

In PD61^Ndi1^, the observed changes in mitochondrial gene expression and gene copy number could be due to increased mitochondrial biogenesis signaling, driven by increased expression of PGC-1α which suggests that retrograde signaling from mitochondria back to the nucleus is increased [90]. Changes in mitochondrial NADH/NAD+ ratios as a result of Ndi1 could underlie the increase in PGC-1α. PGC-1α increases mitochondrial biogenesis as well as enzymes that detoxify ROS and has a global effect on mitochondrial function as reviewed by Austin and St. Pierre [91]. The finding that Ndi1 expression can lead to increased PGC-1α expression is a significant finding worthy of further investigation considering the evidence of reduced biogenesis signaling in PD patients [88]. Therapies that increase PGC-1α have shown some promise in improving mitochondrial function [92,93].

In this study, we anticipated that CLB expression would decline due to increased availability of ATP to fuel autophagic processes in Ndi1-expressing cells. We found significantly reduced levels of Congo red-labeled small protein aggregates in PD61^Ndi1^ (Figure 7), however, the size and number of CLB expressed by PD61^Ndi1^ was not significantly different from PD61. Lack of change in the expression of CLB despite improvements in OXPHOS and reduction in small aggregate density suggests that Ndi1 expression was not sufficient to- (1) prevent CLB formation or (2) induce clearance of CLB. The life cycle of a CLB is not well understood. Puncta of αSYN and Congo red-labeled aggregated proteins are present in nigral neurons in postmortem brain and in PD cybrids [50,94]. The presence of Ndi1 protein in CLB suggests that at least some CLBs were formed after Ndi1 transfection and expression (Figure 8D). This finding indicates that improving ETC function alone is insufficient to prevent CLB formation. Inclusion of protein and gene expression changes as well as autophagy markers in future studies would be informative (see Nashine et al. [95]).

### 4.2. Cellular Consequences of Ndi1 Expression in SH-SY5Y

Unlike the robust changes in PD61^Ndi1^ OCR, OCR in SH-SY5Y^Ndi1^ exhibited only a few altered features. This is consistent with other studies of alternate oxidases like Ndi1. Cannino et al. [36] reported that NDI1 gene expression did not detectably alter cell physiology or mitochondrial function in the absence of ETC dysfunction. Significant changes in SH-SY5Y^Ndi1^ were detected in maximum and spare capacity. Pretreatment of SH-SY5Y cells with rotenone, reduced basal OCR to levels comparable to PD61 (Figure 2C and Figure 3C). In contrast, rotenone pretreatment of SH-SY5Y^Ndi1^ only significantly altered maximum and spare capacity (Figure 3E) confirming that Ndi1 was functionally incorporated into SH-SY5Y mitochondria.

ECAR was significantly decreased (Figure 3B) even though basal OCR in SH-SY5Y was not significantly improved by Ndi1 expression and mitochondrial mass was unaltered in SH-SY5Y^Ndi1^ (Figure 3B). This outcome suggests that SH-SY5Y^Ndi1^ has become more dependent on OCR than SH-SY5Y.

The changes in cell function induced by Ndi1 in SH-SY5Y cells were not as dramatic as the changes seen in PD61. In fact, previous studies have shown that expressing Ndi1 in differentiated neurons did not affect their viability [96]. There was however increased biogenesis signaling via PGC-1α in SH-SY5Y^Ndi1^. At the protein level, the ETC complex assembly was increased in SH-SY5Y^Ndi1^ compared to SH-SY5Y for complexes I, II, III, and IV. This finding was unexpected considering there was no upregulation of mtDNA gene expression or copy number. Previous studies have shown that undifferentiated SH-SY5Y cells have greater dependence on glycolysis than differentiated SH-SY5Y cells [97]. Interestingly, as mentioned above, we observed an increase in maximum capacity in SH-SY5Y^Ndi1^. These data together suggest that SH-SY5Y^Ndi1^ cells could have a higher capacity for OXPHOS compared to the SH-SY5Y cells (Figure 3E), which indicates a shift towards greater dependence on OXPHOS, resulting in the higher levels of ETC complex assembly in SH-SY5Y^Ndi1^.

One drawback to these findings is that we only used two transfected lines in this study, a PD cybrid cell line with typical mitochondrial dysfunction, abnormal mitochondrial morphology, and αSYN aggregation and the parent SH-SY5Y cell line [50]. This preliminary study was designed to test if Ndi1 would be effective at improving mitochondrial function in a sporadic PD model with mitochondrial dysfunction. In light of these findings, this study should be expanded to include cybrid cell lines from PD and other neurodegenerative diseases with mitochondrial dysfunction as well as cybrid lines made from disease-free age-matched control platelets. We expect there will be heterogeneity in the impact of Ndi1 on individual PD cybrid lines, due to the differences in mitochondrial dysfunction among these cell lines [58,59,60].

## 5. Conclusions

In conclusion, the expression of Ndi1 in a trans-mitochondrial cybrid cell model of sporadic PD led to improved mitochondrial function, morphology, and transport. Mitochondrial gene expression, copy number, and biogenesis were also all increased in our PD cybrid cell line after the expression of Ndi1, suggesting that Ndi1 warrants further investigation as a source of insight into potential therapies for PD and other mitochondrial diseases. Although we saw a significant improvement in mitochondrial OCR that exceeded levels in control cybrid cell lines and a reduction in cellular aggregated protein content, Ndi1 expression did not prevent the formation of CLB in the PD cybrid cell line. This finding suggests that supplementing complex I with alternate oxidases like Ndi1 is sufficient to improve mitochondrial function in this PD model, but other additional interventions may be necessary to address the multisystem nature of PD pathogenesis.

## Figures and Tables

**Figure 1 biomolecules-09-00119-f001:**
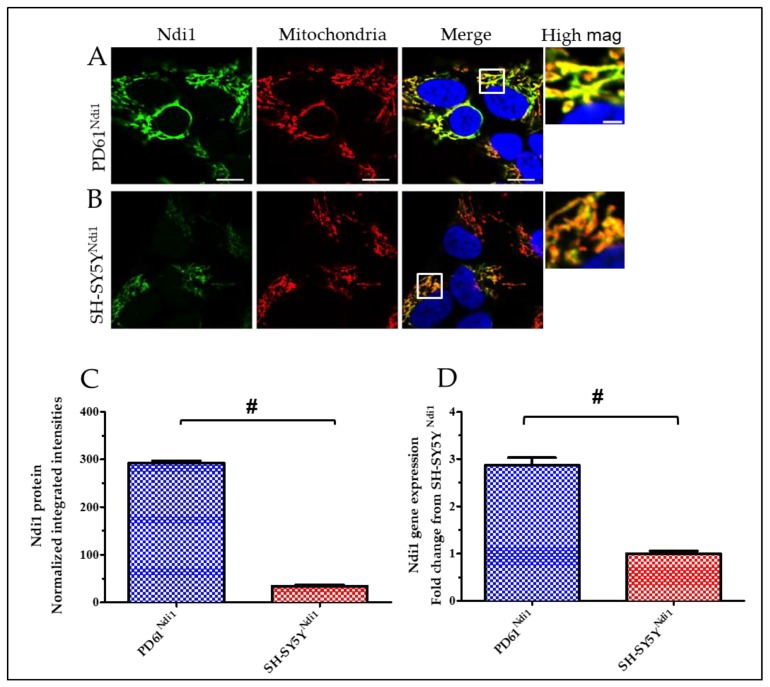
Expression of Ndi1 was greater in PD61^Ndi1^ than in SH-SY5Y^Ndi1^ based on immunofluorescence, Western blot analysis and qRT-PCR. (**A**,**B**) Mitochondria in PD61^Ndi1^ and SH-SY5Y^Ndi1^ cells stained with Ndi1 (green), complex Vα (red) and nuclei labeled with DAPI (blue). Box outline designates an enlarged area of merged image (high mag). Scale bars = 10 μm; high mag scale bar = 2 μm. (**C**) Western blot visualization of Ndi1 protein levels in SH-SY5Y^Ndi1^ and PD61^Ndi1^. Values shown are integrated intensities normalized to actin. (**D**) Quantitative real-time (qRT-PCR) analysis of *NDI1* gene expression levels in PD61^Ndi1^ and SH-SY5Y^Ndi1^ and shown as fold change from SH-SY5Y^Ndi1^ levels. There was an approximately three-fold increase in PD61^Ndi1^. Students *t*-test, n = 3, # *p* < 0.005.

**Figure 2 biomolecules-09-00119-f002:**
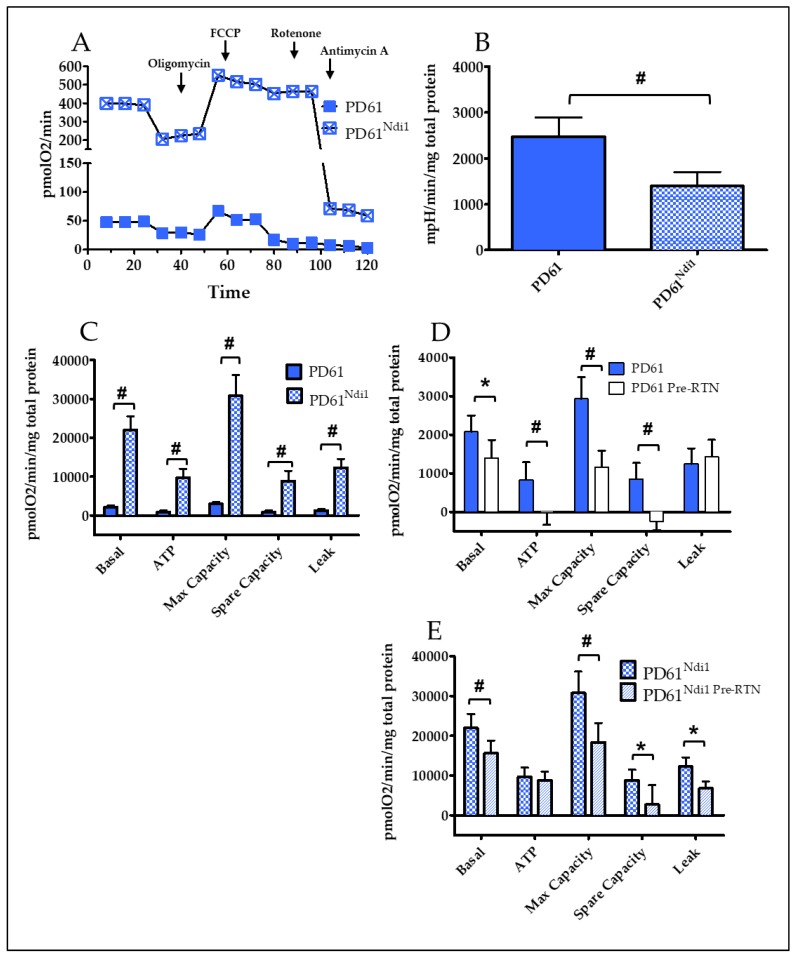
PD61^Ndi1^ exhibited improved mitochondrial respiration. (**A**) Representative traces of oxygen consumption rates (OCR) from PD61 and PD61^Ndi1^. Indicated inhibitors were added sequentially. (**B**) ECAR (glycolysis) rates in PD61 and PD61^Ndi1^. (**C**) Normalized basal, ATP-linked respiration, max capacity, spare capacity, and leak rates for PD61 and PD61^Ndi1^. (**D**,**E**) Normalized basal respiration rates from rotenone (−) and rotenone (+)/Pre-RTN experiments using PD61 (D) and PD61^Ndi1^ (E). Rotenone (−) experiments were performed using normal running media. Rotenone (+)/Pre-RTN experiments (D,E) were performed with a 30 nM rotenone pretreatment to inhibit endogenous complex I. Two-way ANOVA with Bonferroni post-hoc analysis, n = 10, * *p* < 0.01, # *p* < 0.001.

**Figure 3 biomolecules-09-00119-f003:**
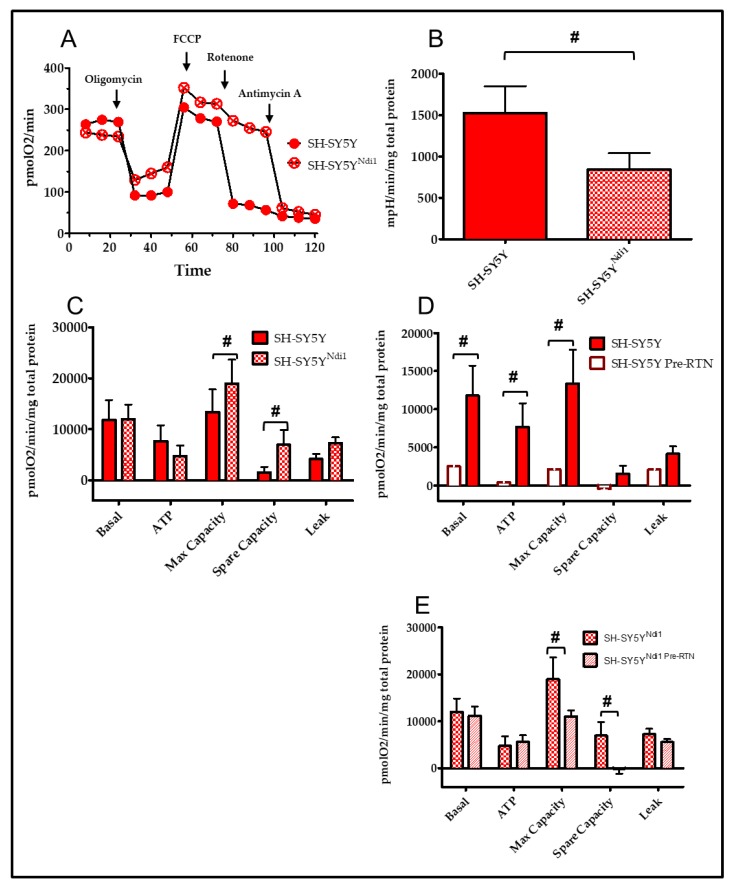
Ndi1 expression in SH-SY5Y^Ndi1^ improved mitochondrial respiration. (**A**) Representative traces of oxygen consumption rates (OCR) from SH-SY5Y and Sh-SY5Y^Ndi1^. Indicated inhibitors were added sequentially. (**B**) ECAR (glycolysis) rates in SH-SY5Y and SH-SY5Y^Ndi1^. (**C**) Normalized basal, ATP-linked respiration, max capacity, spare capacity and leak rates for PD61 and PD61^Ndi1^. (**D**,**E**) Normalized basal respiration rates from rotenone (−) and rotenone (+)/Pre-RTN experiments using SH-SY5Y (D) and SH-SY5Y^Ndi1^ (**E**). Rotenone (−) experiments were performed using normal running media. Rotenone (+)/Pre-RTN experiments (**D**,**E**) were performed with a 30 nM rotenone pretreatment to inhibit endogenous complex I. Two-way ANOVA with Bonferroni post-hoc analysis, n = 10, # *p* < 0.001.

**Figure 4 biomolecules-09-00119-f004:**
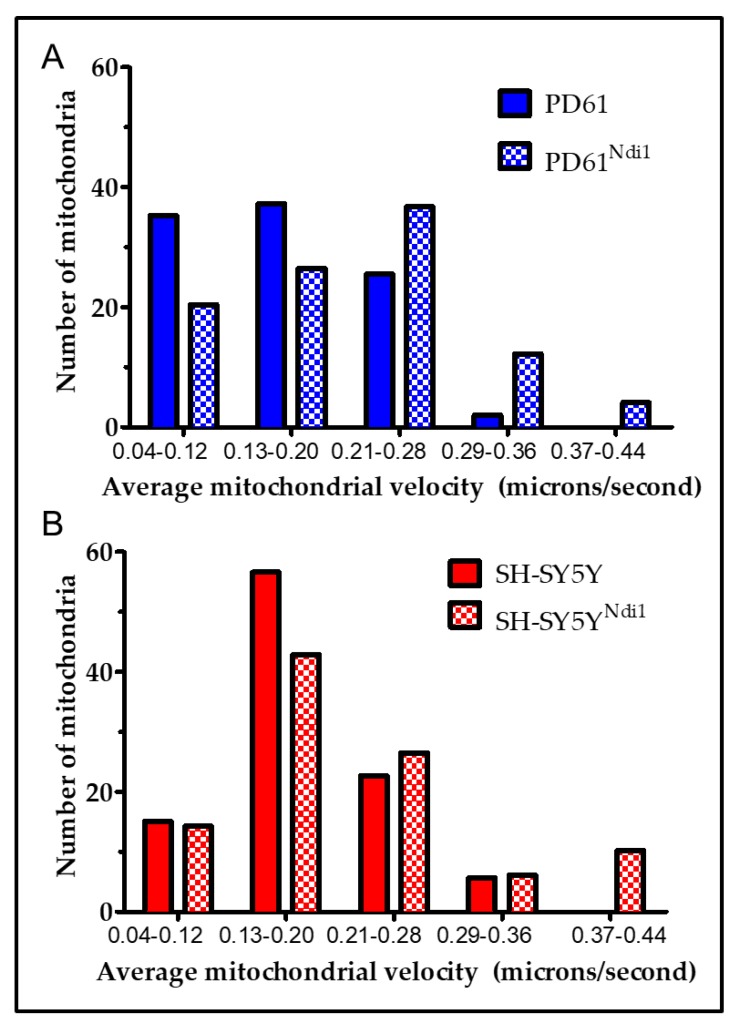
Mitochondrial movement by axonal transport is altered in PD^Ndi1^. (**A**) Histogram of average mitochondrial velocity (PD61- 8–50 mitochondria/process in 39 processes from four different cultures; PD61^Ndi1^ 9–44 mitochondria/process in 36 processes from six different cultures). (**B**) Histogram of average mitochondrial velocity (SH-SY5Y 8–46 mitochondria/process in 59 different processes in four different cultures; SH-SY5Y^Ndi1^ 5–34 mitochondria/processes in 59 different processes from five different cultures).

**Figure 5 biomolecules-09-00119-f005:**
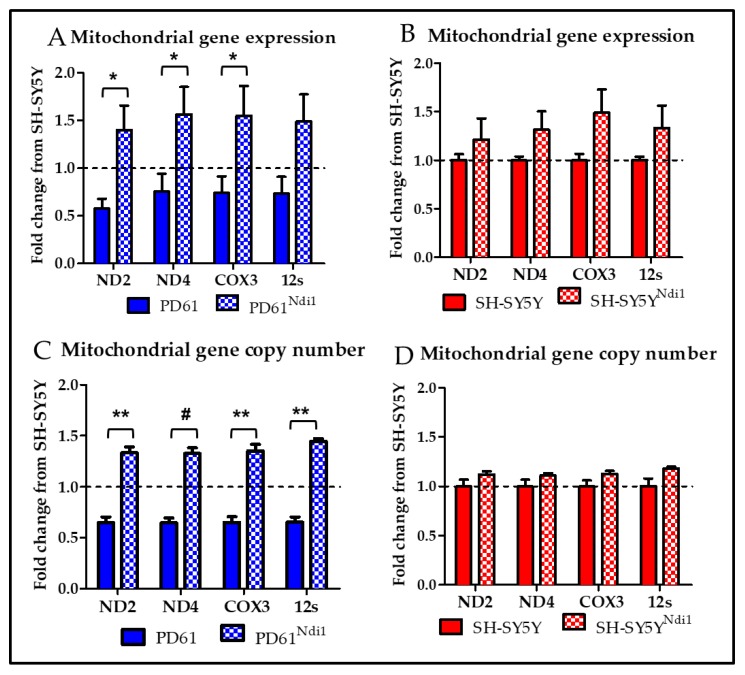
Mitochondrial gene expression and copy number after Ndi1 expression. qRT-PCR analysis of mtDNA-encoded gene expression is shown as fold change from SH-SY5Y levels (dashed line at 1.0). (**A**) Gene expression was significantly increased for ND2, ND4, and COX3 in PD61^Ndi1^ compared to PD61. (**B**) mtDNA-encoded gene expression was not significantly changed in SH-SY5Y^Ndi1^ compared to SH-SY5Y. qRT-PCR analysis of mtDNA-encoded gene copy number shown as fold change from SH- SY5Y levels (dashed line at 1.0). (**C**) Gene copy number was elevated in all four genes measured (ND2, ND4, COX3, and 12s) in PD61^Ndi1^ compared to PD61. (**D**) There was no change in mtDNA-encoded copy number in SH-SY5Y^Ndi1^ relative to SH-SY5Y. Student’s *t*-test, n = 3, * *p* < 0.05, ** *p* < 0.01, # *p* < 0.005.

**Figure 6 biomolecules-09-00119-f006:**
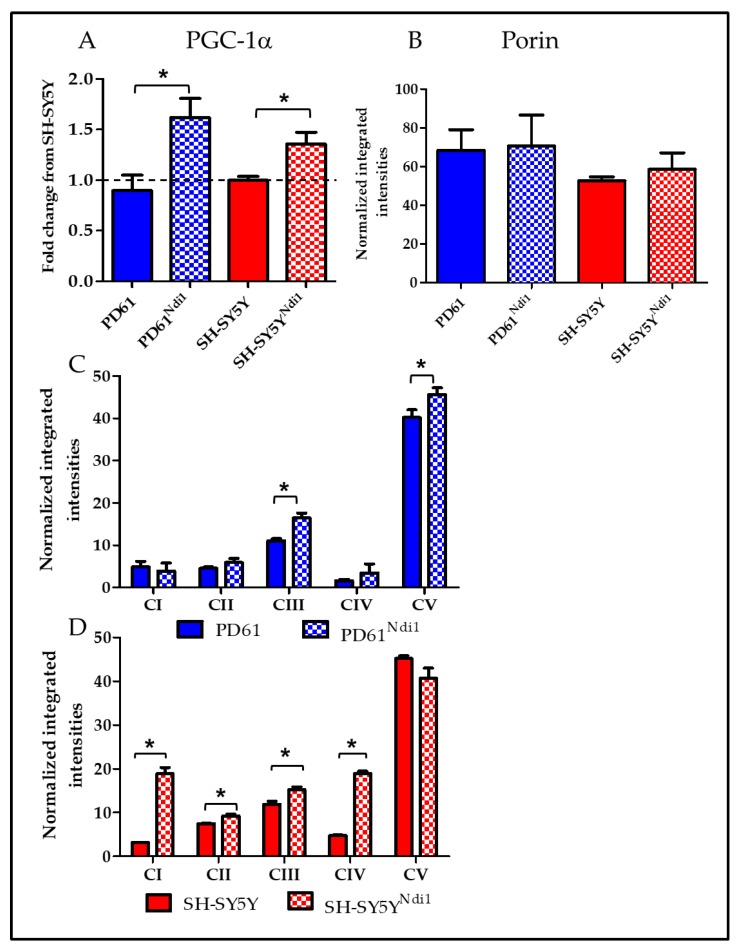
Mitochondrial biogenesis and electron transport chain assembly after Ndi1 expression. (**A**) qRT-PCR analysis shows that PGC-1α is increased in PD61^Ndi1^ relative to PD61 and in SH-SY5Y^Ndi1^ versus SH-SY5Y. Dashed line represents SH-SY5Y mean. Student’s *t*-test, n = 3, * *p* < 0.05. (**B**) qRT-PCR analysis shows that porin is not significantly different in PD61, PD61^Ndi1^, SH-SY5Y or SH-SY5Y^Ndi1^. (**C**,**D**) Graphs showing normalized integrated intensities of ETC complex assembly for all five ETC proteins, normalized to porin (**B**). (**C**) ETC complex assembly was increased in complexes III and V in PD61^Ndi1^ compared to PD61. (**D**) ETC assembly was increased in complexes I-IV for SH-SY5Y^Ndi1^ compared to SH-SY5Y. Student’s *t*-test, n = 3 for SY5Yand SY5Y^Ndi1^, n = 4 for PD61 and PD61^Ndi1^, * *p* < 0.05.

**Figure 7 biomolecules-09-00119-f007:**
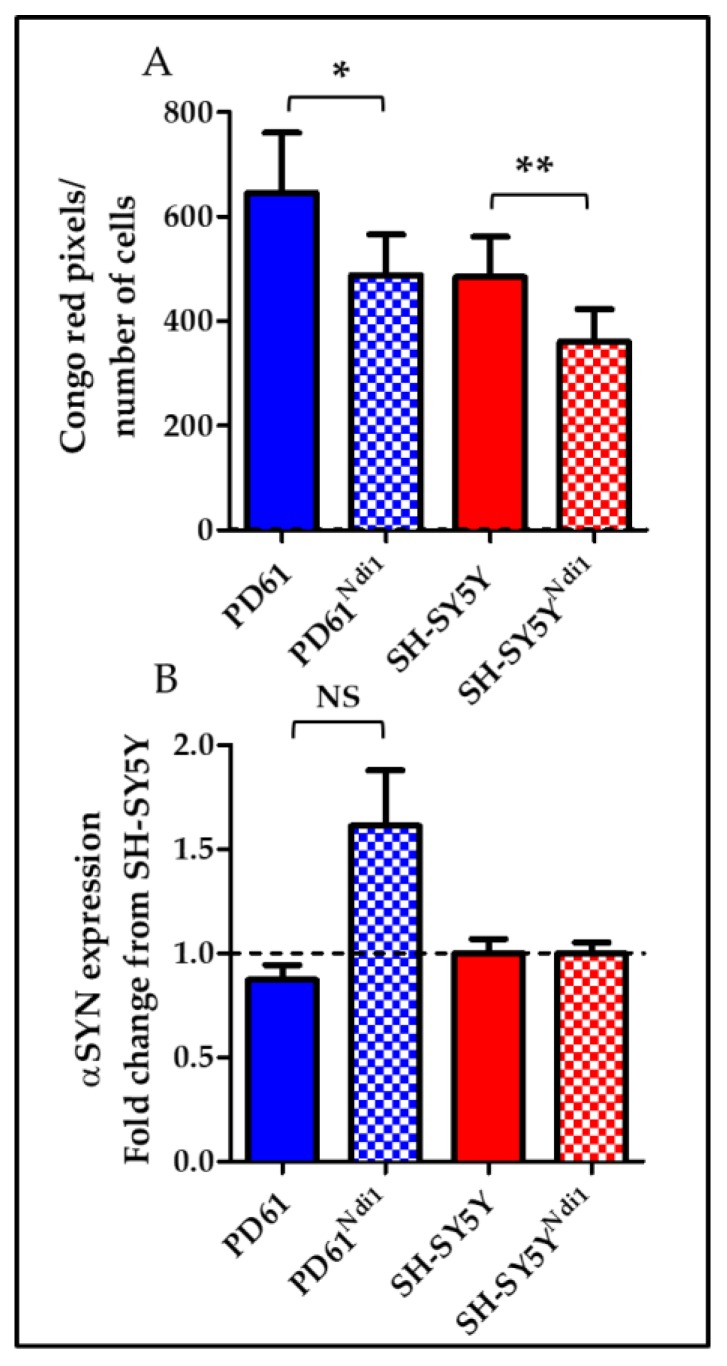
Aggregated protein and αSYN expression gene expression. (**A**) Congo red pixels were counted and normalized to cell number as a measure of cellular aggregated protein content. Both SH-SY5Y^Ndi1^ and PD61^Ndi1^ showed a decrease in total aggregated protein content after Ndi1 expression. Student’s *t*-test, n = 3. * *p* < 0.05, ** *p* < 0.01 (**B**) qRT-PCR for αSYN showed no significant change in αSYN expression after Ndi1 expression in SH-SY5Y^Ndi1^. PD61^Ndi1^ shows a trend towards increased αSYN gene expression (*p* = 0.055). Student’s *t*-test, n = 3.

**Figure 8 biomolecules-09-00119-f008:**
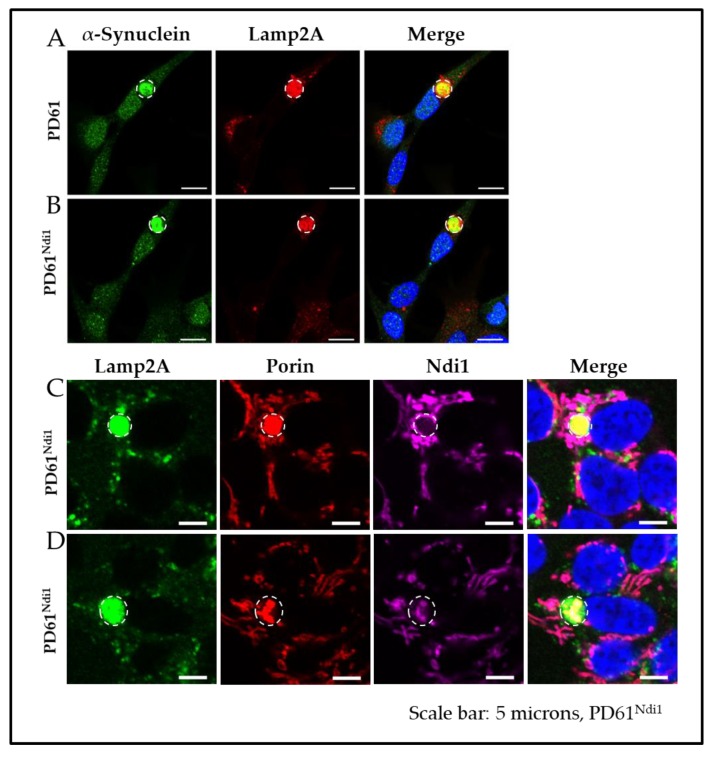
Cybrid Lewy body expression after *NDI1* gene expression. (**A**,**B**) Representative CLB from PD61 and PD61^Ndi1^ showing colocalization of αSYN with Lamp2A (lysosome-associated membrane protein). Scale bars = 10 μm (**C**,**D**) Representative images from PD61^Ndi1^ showing CLB present in cells expressing Ndi1, Lamp2A and porin. Scale bars = 5 μm.

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
