# Peer review of "Differential Effects of Yeast NADH Dehydrogenase (Ndi1) Expression on Mitochondrial Function and Inclusion Formation in a Cell Culture Model of Sporadic Parkinson’s Disease"

_biomolecules, 2019, doi:10.3390/biom9040119_

Round 1
Reviewer 1 Report
Manuscript: Biomolecules 450109
Title: Differential effects of yeast NADH dehydrogenase 2 (Ndi1) expression on mitochondrial function and 3 inclusion formation in a cell culture model of 4 sporadic Parkinson’s disease
Authors: Emily N. Cronin-Furman, Jennifer Barber-Singh, Kristen E. Bergquist, Takao Yagi, Patricia A. Trimmer
General Comments: This is a very well-written manuscript that uses the model of SH-SY5Y cybrids containing mitochondria from a well-characterized Parkinson’ disease patient to determine if transfection of Ndi1 can rescue mitochondrial function and reduce the formation of Lewy body in vitro. The methods are diverse and appropriate for the study. The results and interpretations are clearly presented and clinically important. One of the most useful parts of the manuscript is that they show improvement of the mitochondrial dysfunction does not correlate with reduction of the synuclein/Lewy body formation. This will guide future studies to look for drugs/techniques that can improve both problems associated with Parkinson’s disease (mitochondrial damage and Lewy body production).
Specific Comments:
1. The legend for Figure 6 (lines 442-447) is not consistent with the text (lines 451-460) or figure. Please correct the so it is accurate.
2. Methods. For those not familiar with the cybrid model, can the authors state how the SH-SY5Y cells were made Rho0? How long did it take to create the PD-61 cybrid? What is the passage used for the studies/
3. Line 466: should be Figure 7B not 6B
4. Please check the references as some have full titles for journals while others are abbreviated. Some are capitalized but others are not. Also the formats are not consistent.
Author Response
Response to Reviewer 1 comments
1) The legend for Figure 6 (lines 442-447) is not consistent with the text (lines 451-460) or figure. Please correct the so it is accurate.
The text for Figure 6 has been corrected.
2. Methods. For those not familiar with the cybrid model, can the authors state how the SH-SY5Y cells were made Rho0? How long did it take to create the PD-61 cybrid? What is the passage used for the studies/
A brief description of how the cybrid line was made has been included in the methods section.
3. Line 466: should be Figure 7B not 6B
Line 466 was corrected.
4. Please check the references as some have full titles for journals while others are abbreviated. Some are capitalized but others are not. Also the formats are not consistent.
The titles were corrected for capitalization. The format of the reference section was generated by Endnote using the style for Biomolecules.
Reviewer 2 Report
The paper outlines the expression of complex I from yeast in a sPD and control cybrid line. The generation of the cybrids and their accompanying abnormalities have been published before. The authors compare the Ndi transduced cell line from sPD patient and Ndi expressing line from a control to their parental lines. The authors do not have a parental line for comparison which has undergone the same transduction procedure as the Ndi lines; therefore any changes they see in the lines expressing Ndi they cannot rule out they are not artifacts of the transduction process. Furthermore the authors do not compare the sPD line to the control line. It would be useful to see if the mitochondrial parameters are improved to the level of a control.
With regard the cybrid Lewy Body expression it would be useful to know how long the cybrids had been expressing Ndi before the measurements were made.
Author Response
Response to Reviewer 2 comments
1) The authors compare the Ndi transduced cell line from sPD patient and Ndi expressing line from a control to their parental lines. The authors do not have a parental line for comparison which has undergone the same transduction procedure as the Ndi lines; therefore any changes they see in the lines expressing Ndi they cannot rule out they are not artifacts of the transduction process.
The reviewer is correct in pointing out that we did not use a control vector in any cell line in this study. Cell lines expressing a control vector were not utilized for this study in keeping with previous Ndi1 expression studies in cells by T. Yagi and other labs. Gene delivery by adeno-associated virus (AAV) is safe, effective and elicits long-lasting expression and has not been associated with any of the mitochondrial or cellular changes reported in this manuscript.
2) Furthermore, the authors do not compare the sPD line to the control line. It would be useful to see if the mitochondrial parameters are improved to the level of a control.
As we admitted in the manuscript, we were not able include a control cybrid cell line that was transfected for Ndi1. We've included a statement in section 3.3 comparing the basal and max capacity respiration in PD61Ndi1 to the mean OCR from four control cybrid lines. The conclusion was also amended.
3) With regard the cybrid Lewy Body expression it would be useful to know how long the cybrids had been expressing Ndi1 before the measurements were made.
A statement was also added in the methods describing the passage numbers for PD and SH-SY5Y cells. Cells were passed roughly on a weekly basis. Frozen stocks were thawed every 2 months.